# An Internal-Electrostatic-Field-Boosted Self-Powered Ultraviolet Photodetector

**DOI:** 10.3390/nano12183200

**Published:** 2022-09-15

**Authors:** Dingcheng Yuan, Lingyu Wan, Haiming Zhang, Jiang Jiang, Boxun Liu, Yongsheng Li, Zihan Su, Junyi Zhai

**Affiliations:** 1Center on Nanoenergy Research, Guangxi Key Laboratory for Relativistic Astrophysics, School of Physical Science and Technology, Guangxi University, Nanning 530004, China; 2CAS Center for Excellence in Nanoscience, Beijing Key Laboratory of Micro-Nano Energy and Sensor, Beijing Institute of Nanoenergy and Nanosystems, Chinese Academy of Sciences, Beijing 100083, China

**Keywords:** self-powered photodetector, the photovoltaic effect, the electrostatic potential

## Abstract

Self-powered photodetectors are of significance for the development of low-energy-consumption and environment-friendly Internet of Things. The performance of semiconductor-based self-powered photodetectors is limited by the low quality of junctions. Here, a novel strategy was proposed for developing high-performance self-powered photodetectors with boosted electrostatic potential. The proposed self-powered ultraviolet (UV) photodetector consisted of an indium tin oxide and titanium dioxide (ITO/TiO_2_) heterojunction and an electret film (poly tetra fluoroethylene, PTFE). The PTFE layer introduces a built-in electrostatic field to highly enhance the photovoltaic effect, and its high internal resistance greatly reduces the dark current, and thus remarkable performances were achieved. The self-powered UV photodetector with PTFE demonstrated an extremely high on–off ratio of 2.49 × 10^5^, a responsivity of 76.87 mA/W, a response rise time of 7.44 ms, and a decay time of 3.75 ms. Furthermore, the device exhibited exceptional stability from room temperature to 70 °C. Compared with the conventional ITO/TiO_2_ heterojunction without the PTFE layer, the photoresponse of the detector improved by 442-fold, and the light–dark ratio was increased by 8.40 × 10^5^ times. In addition, the detector is simple, easy to fabricate, and low cost. Therefore, it can be used on a large scale. The electrostatic modulation effect is universal for various types of semiconductor junctions and is expected to inspire more innovative applications in optoelectronic and microelectronic devices.

## 1. Introduction

With the rapid development of the Internet of Things (IoT), ensuring a large-scale distributed power supply has become a challenge because of the numerous sensors and detectors used in IoT systems. Conventional power sources, such as batteries, have limited life and cause environmental pollution. Furthermore, it is difficult to manage, replace, and maintain power sources. The self-powered device without an external power supply is a promising solution for the development of a green and sustainable IoT [1,2], which has attracted considerable attention.

Ultraviolet (UV) photodetectors, as basic units of photoelectric information systems and IoT, are widely used in fields such as fire warning, astronomical exploration, environmental monitoring, chemical/biological sensing, and optoelectronic storage [3,4,5,6,7]. Generally, most UV photodetectors, such as UV phototubes, and semiconductor-based diodes [8,9], require external power sources for operation. For example, the working voltage of the ultraviolet phototube reached as high as one hundred volts. Although a p-n junction device can function without a power source, its dark current is high, and the detection performance is limited. Therefore, a reverse bias is typical for the p-n junction photodetectors. Currently, self-powered ultraviolet photodetectors have attracted considerable interest because of their great potential for energy conservation and environmental protection. Self-powered ultraviolet photodetectors can be mainly categorized into three types, namely semiconductor-based diodes, photochemical batteries, and devices with triboelectric nanogenerators (TENGs) as power sources. Among them, semiconductor junction devices, such as the Schottky junction [10,11,12], p-n junction [13,14,15], and heterojunction detectors [16,17,18,19], which are based on the photovoltaic effect, are typically used for self-powered devices. Numerous studies have focused on developing junction-based self-powered photodetectors using various UV materials, such as zinc oxide [20], titanium dioxide [21], gallium oxide [22,23,24,25,26], and perovskite [27,28]. However, the difficult preparation of high-quality semiconductor junctions restricted their self-powered detection performance, and large-scale applications remain a challenge. In terms of the other two types of UV self-powered photodetector, the photochemical cells faced the shortcomings of limited working time, corrosiveness, and an unfriendly environment. Moreover, a photodetector with TENG actually worked under a bias voltage [29]. A TENG collects the mechanical energy of the surrounding environment and converts it into electrical energy to supply power to the device. The detector was not self-driving, but a TENG functioned served as a power source. Furthermore, the TENG required a certain moving mode to generate electricity, which limited its application.

To further improve the performance of self-powered detectors, coupling other effects to enhance the photovoltaic effect is an effective approach. For example, the pyroelectric effect [30,31], localized surface plasmon resonance effect [32,33], piezoelectric effect [34,35] and the self-polarization effect [36], are used to develop the enhanced self-powered photodetectors. However, these enhanced photodetectors still required high-quality semiconductor junctions. Self-powered photodetectors with ultra-low dark current and high photocurrent that do not require high-quality junctions are very scarce. Therefore, new strategies and techniques are urgently needed to develop self-powered photodetectors with high performance and low cost.

In this work, an electret was creatively used to enhance the photovoltaic effect, and a self-powered UV photodetector with remarkable performances was fabricated. The internal-electrostatic-field-boosted self-powered (IEFB-SP) photodetector consisted of a semiconductor heterojunction and an electret layer. In our IEFB-SP PD, there exist two built-in electric fields. One is on the ITO/TiO_2_ interface, and the other is on the TiO_2_/PTFE interface. ITO/TiO_2_ is known as a semiconductor heterojunction that produces a built-in electric field at the junction region due to the difference in the work function of the two materials. On the TiO_2_/PTFE interface, there exists a built-in electrostatic field because PTFE molecules with strong electronegativity gain electrons from TiO_2_ molecules. The electrostatic field generated by the negatively charged PTFE layer is in the same direction as the built-in electric field on ITO/TiO_2_ interface, and the two electric fields are in series. Under UV light irradiation, TiO_2_ absorbs UV photons and generates electron–hole pairs. The electrostatic field boosts the separation of electron–hole pairs and pushes the electrons to ITO while the holes to PTFE. With the coupling effect of the built-in electric field and the electrostatic field, an enhanced photovoltaic effect is achieved. The electrostatic potential generated on the electret interface accelerated the separation and diffusion of photo-generated carriers, which highly enhanced the photovoltaic effect, and an excellent self-powered photodetection was achieved. Under the zero-bias voltage, the response of the device to a 365-nm ultraviolet light was as high as 76.87 mA/W, its specific detectivity was 4.79 × 10^12^ jones, and the external quantum efficiency reached 26.48%. Compared with a conventional semiconductor heterojunction device without an electret layer, the photoresponse of the IEFB-SP UV detector was increased by 442 times, and the light–dark ratio was increased by 8.40 × 10^5^ times. Furthermore, the proposed device was easy to manufacture, simple, and environment friendly, and could be easily used on a large scale. More importantly, the electrostatic field of an electret is universal for modulating carrier transport properties for various types of semiconductor junction devices. It provides an innovative approach for the development of low-cost, high-performance self-powered photodetectors. 

## 2. Materials and Methods

**Device fabrication:** For the preparation of the TiO_2_ film: Commercially available ITO glass, which was purchased from (Xiang Nan Science & Technology Co., Ltd., Hunan, China). and was used as the substrate, washed with acetone, ethanol, and deionized water in an ultrasonic cleaner for 30 min, and then placed in a drying oven at 60 °C. Next, a magnetron sputtering system was used to grow a TiO_2_ film on the ITO surface. During the growth process, the substrate temperature was 200 °C, the sputtering time was 120 min, the flow rate of Ar was 80 sccm, the RF sputtering power was 150 W, and the working pressure was 0.77 Pa. Finally, the sample was annealed at 550 °C in the air for 90 min to form a TiO_2_ film.

**Preparation of the PTFE layer:** A magnetron sputtering system was used to grow a PTFE film on the surface of TiO_2_. During the growth process, the substrate temperature was normal, the sputtering time was 30 min, the Ar gas flow was 80 sccm, the sputtering power was 100 W, and the working pressure was 0.7 Pa. Finally, a high-vacuum thermal evaporation system was used to vapor-deposit Cu (150 nm) electrodes on PTFE.

**Material characterizations:** SEM was performed (sigma 500, Zeiss, Oberkochen, Germany) to measure the surface morphologies of TiO_2_, PTFE films, and the cross-section image of the ITO/TiO_2_/PTFE film. The crystalline quality of the TiO_2_ film was measured by using an X-ray diffractometer (SmartLab3KW, RIGAKU, Tokyo, Japan). A UV spectrophotometer (TU-1901, PERSEE, Beijing, China) was used to measure the optical transmission spectra. AFM was performed (Dimension Icon, BRUKER NANO INC, Karlsruhe, Germany) to measure the surface roughness of the TiO_2_ film.

**Photodetection performance measurements:** The photoelectric performances of the device were measured by using an electrometer (Keithley Model 6514, OH, USA) and an oscilloscope (Tectronix MDO 3012, OH, USA). The responsivity and EQE measurements were performed by using a testing system of the photodetector (DSR-3110-PZ, Zolix, Beijing, China) with a UV-enhanced light source. The ultraviolet light source (WarSun R838) had a wavelength of 365 nm and an intensity of 15.94 mW/cm^2^, which was measured by a high-precision photoelectric laser power meter (PS100, CNI, Changchun, China).

## 3. Results and Discussion

### 3.1. Device Structure and Working Mechanism

Figure 1a displays the design of the proposed IEFB-SP ultraviolet photodetector. The photodetector was a four-layer film of ITO/TiO_2_/PTFE/Cu. Among them, ITO/TiO_2_ functioned as a conventional n-type heterojunction. The PTFE was an electret layer that generated static electricity on the interface and also functioned as a high-resistance layer to reduce the dark current. ITO and Cu were used as the electrodes of the detector. The working principle of the device is displayed in Figure 1b,c. On the one hand, a difference in electron affinities between ITO and TiO_2_ produces a built-in electric field in the ITO/TiO_2_ junction. On the other hand, the PTFE, owning high electronegativity, gains electrons from TiO_2,_ and a built-in electrostatic potential generates on the TiO_2_/PTFE interface. Based on the first principles calculation and molecular dynamics simulation, the intermolecular transferred charges on the TiO_2_/PTFE and the TiO_2_/PDMS interface were quantitatively analyzed (see Appendix A), and the varying contact electrification properties with intermolecular distances are shown in Appendix A, revealing an existing electrostatic field on the TiO_2_/PTFE and TiO_2_/PDMS interface. Experimentally, based on the coupling of contact electrification and electrostatic induction, a triboelectric nanogenerator (TENG) composed of TiO_2_ and PTFE was fabricated, and the charge transfer properties between the TiO_2_ and PTFE surfaces were investigated when they were in contact. As shown in Appendix A, the surface of PTFE is negatively charged, and the surface of TiO_2_ is positively charged after contact. Without UV lights, photo-generated carriers were not developed in the TiO_2_ layer, and the device outputted a small short-circuit dark current (Figure 1d) because of a weak contact potential difference of ITO/TiO_2_ heterojunction and the high impedance of the PTFE layer. With UV illumination, photo-generated carriers were produced in TiO_2_. Under the simultaneous actions of the built-in electric field of ITO/TiO_2_ heterojunction and the electrostatic field of TiO_2_/PTFE interface, photo-generated carriers were accelerated to separate and diffuse to generate a boosted photovoltaic effect. The negative potential of PTFE remarkably promoted the optical response and considerably improved output performances. The equivalent circuit diagrams of the detector in light-off and light-on states are illustrated in Figure 1e,f.

### 3.2. Material Characterizations

Figure 2a displays the scanning electron microscopy (SEM) images of the ITO/TiO_2_/PTFE/Cu film and the photograph of the fabricated device. The thickness of the TiO_2_ film was 236 nm, and that of the PTFE film was 150 nm. The optical transmission spectra of ITO with the glass substrate and the sample after the growth of the TiO_2_ film are displayed in Figure 2d. The cut-off absorption wavelength of ITO was 303 nm. After the growth of the TiO_2_ film, the cut-off absorption edge was red-shifted at 344 nm. The bandgap of the ITO film was 3.75 eV, and the bandgap of the TiO_2_ film was 3.44 eV, which was calculated by the optical absorption spectra. Figure 2e displays the X-ray diffraction pattern of the TiO_2_ film. Taking the XRD standard card of PDF #89-4920 and PDF #21-1272 as a reference, the prepared TiO_2_ film was a mixture of anatase and rutile. Figure 2b,c display the surface SEM images of the TiO_2_ and PTFE films. The atomic force microscopy (AFM) images of the TiO_2_ surface are displayed in Figure 2f. The root mean square roughness of the TiO_2_ surface, average roughness, and maximum roughness were 2.26, 1.81, and 15.5 nm, respectively. Certain surface roughness of TiO_2_ was conducive to the generation of the electrostatic field in the interface between TiO_2_ and PTFE films.

### 3.3. Photodetection Performance of IEFB-SP Device

The ultraviolet detection performance of the IEFB-SP device at 0 V is displayed in Figure 3. A UV light-emitting diode (LED) with a wavelength of 365 nm fixed on a linear motor periodically illuminated the detector by linear motion, producing turn-on and turn-off states (see Appendix A). In the measurements, the humidity of the air environment was between 30 RH and 40 RH, and the UV LED moved to illuminate the device and stayed for 3 s and then moved away at a speed of 1 m/s. The proposed device exhibited excellent self-powered photodetection performances, as displayed in Figure 3. Without UV light, the dark current of the device was extremely low, 8.03 × 10^−12^ A. With UV light, due to the high resistance of the PTFE layer, the dark current (I_d_) of the IEFB-SP device was extremely low, 8.03 × 10^−12^ A (see Figure 3a). With UV irradiation, the photocurrent (I_p_) was approximately 2.00 × 10^−6^ A, and the light-to-dark current ratio reached 2.49 × 10^5^. The current-voltage (I-V) curves (Appendix A) show an obvious increase in the magnitude of photocurrent at zero bias in the light state compared with the dark state. As illustrated in Figure 2b. the photovoltage (V_p_) and dark voltage (V_d_) of the IEFB-SP device were 0.02 V and 8.52 × 10^−5^ V, respectively. To figure out the functions of the PTFE layer, we measured the internal resistances for photodetectors using the impedance matching method and the photodetection performance of a conventional ITO/TiO_2_/Cu heterojunction without a PTFE layer for comparison, as displayed in the Appendix A (see Appendix A), and the performance comparisons are listed in Appendix A. We can see that, without PTFE, the average photocurrent (I_p_) and dark current (I_d_) of the ITO/TiO_2_/Cu device at 0 V are 4.76 × 10^−6^ A and 3.67 × 10^−6^ A, respectively, and the on–off ratio is only 1.30. Connecting the TiO_2_ and Cu, the I–V curve (see Appendix A) reveals there is an excellent ohmic contact between the Cu electrode and the TiO_2_, and the main contact potential difference was generated by the ITO/TiO_2_ interface. In contrast, the light–dark ratio of the device with PTFE was increased by 8.40 × 10^5^ times. The photovoltage increment (V_p_–V_d_) is increased by 665 times, as presented in Appendix A. It is interesting that the internal resistance of the device with PTFE increased by ~ 1.3 × 10^5^ times, but the photocurrent only decreased by 2.4 times. This is because the strong built-in electrostatic field significantly improved the photocurrent. Therefore, the PTFE has double functions. One is the high internal resistance of PTFE, which greatly reduces the dark current, and the other is the built-in electrostatic field, which remarkably enhances the photovoltaic effect. Thus, the IEFB-SP device achieved high performance of low dark current and large photocurrent.

To further quantify the photocurrent properties enhanced by the PTFE, the transferred charge was measured, as displayed in Figure 3c. As the UV light periodically irradiated on the device, the transferred charge increased stepwise with the irradiation times, and the amount of transferred charge for each illumination was 5.97 μC. The current calculated by I = q/t was 1.99 × 10^−6^ A, which is consistent with the detection current. As for the device without PTFE, the amount of transferred charge in a light switching cycle is 14.08 μC, as shown in Appendix A. When we reversed the positive and negative connections between the ITO and Cu electrodes, the current value remained unchanged, but the current direction was opposed (see Appendix A), indicating that the measured signals were produced by the absorbed lights. 

Figure 3d displays the photocurrent dependence on the UV intensity in the intensity range of 0.29–15.94 mW/cm^2^. The photocurrent increased gradually with the UV power density. With a low optical power density of 0.29 mW/cm^2^, the device still exhibited a superior response. The responsivity R and the detectivity D* of the self-powered photodetector were evaluated using the following expression [37,38]:(1)$$R=\frac{I_{P}-I_{D}}{P},$$(2)$$D^{*}=\frac{\left(I_{P}-I_{D}\right)\sqrt{A}}{P\sqrt{2{\mathrm{eI}}_{D}}},$$
where I_P_ is the photocurrent, I_D_ is the dark current of the device, P is the incident optical power, A is the effective detection area, and e is the amount of electronic charge. At 365-nm wavelength, the incident light power was 0.64 mW/cm^2^, and the beam size was 1 mm^2^. The responsibility and the detectivity of the IEFB-SP photodetector were 72.41 mA/W and 4.51 × 10^12^ jones, respectively. The photocurrent and responsivity as a function of illumination intensity are displayed in Figure 3e. The photocurrent followed the power law [39,40] of I_P_ ∝ P^β^, where the β value is 1.03, indicating the photocurrent was almost linearly increased with the incident optical power. The responsivity exhibited excellent stability with the increase in the optical power.

Figure 3f displays the responsivity of the IEFB-SP device at various light wavelengths. As the optical wavelengths were 325, 365, 396, 457, 532, and 607 nm, the responsivity of the device was 19.10, 72.24, 28.93, 14.14, 4.20, and 2.35 mA/W, respectively. Without regard to the absorption of the ITO layer in the UV region, the device exhibits a good UV/visible rejection ratio. A photodetector testing system was used to reveal the spectral response characteristics of the device (ITO/TiO_2_/PTFE/Cu) and the conventional heterojunction (ITO/TiO_2_/Cu) in the wavelength range of 250–700 nm (see Appendix A) in the Appendix A. At the UV wavelength of 360 nm, the responsivity of our device reached as high as 76.874 mA/W, while that of the device without PTFE was only 0.174 mA/W. The responsivity increased by 442 times.

The external quantum efficiency (EQE) of the photodetector is another key parameter for evaluating the photodetection performance and is expressed as follows [41,42]:(3)$$\mathrm{EQE}=\frac{\mathrm{hcR}}{e\lambda },$$
where h represents the Planck constant, c is the velocity of the incident light, e is the charge, and λ is the wavelength of the incident light. The EQE was 24.54% for the proposed device at a wavelength of 365 nm. Under the illumination of a xenon lamp of a photodetector testing system, the EQEs of our detector (ITO/TiO_2_/PTFE/Cu) and the conventional heterojunction (ITO/TiO_2_/Cu) in the wavelength range of 250–700 nm are displayed in Appendix A. As the ultraviolet wavelength is 360 nm, the EQE of our device reached as high as 26.48%, whereas that of the conventional heterojunction device without PTFE was 0.06%.

We evaluated the response speed of the IEFB-SP device. The rise time (T_R90_) is defined as the time required by the photovoltage to reach 90% of the maximum photovoltage, and the decay time (T_D10_) is the time at which the photovoltage decreases to 10% of the maximum photovoltage. The rise and decay times of this photodetector were 7.44 and 3.75 ms, respectively (see Figure 4a,b), and it had a fast response. The rise and fall times of the conventional ITO/TiO_2_/Cu heterojunction were 0.83 s and 1 s, respectively, as displayed in Appendix A, which is slower by 111 and 261 times, respectively, than those of the device with PTFE.

Furthermore, we assessed the temperature stability and environmental stability of the IEFB-SP device. Figure 4c and Appendix A exhibited the photocurrent and dark current characteristics at various temperatures from room temperature to 70 °C, respectively. With the increase in the temperature, the photocurrent and dark current both increased slightly, but the light-to-dark ratio remained unchanged. Compared with the conventional photodetector of the semiconductor heterojunction, it exhibited exceptionally superior temperature stability. Furthermore, we irradiated the device continuously for 2 h and tested it repeatedly for 96 circles, and its photocurrent was invariable, as displayed in Appendix A. After placing in an air environment for 60 days, the IEFB-SP photodetector with PTFE maintained its excellent photodetection performance (see Figure 4d).

### 3.4. Influence of Dielectric Materials

We studied the influence of dielectric layer thickness and material on the photodetection performances of IEFB-SP devices. Figure 4e displays the photocurrent properties of the device with PTFE films of thicknesses 50, 150, 300, and 800 nm. The thinner the thickness of PTFE was, the larger photocurrent and dark current were. Thicker PTFE films not only increased the internal resistance but also reduced the effect of electrostatic induction, which resulted in a considerable decrease in the photocurrent. As the thickness increased, the voltage increased (Appendix A). Among the four thicknesses of PTFE membranes, the detector with a thickness of 150 nm presented the best photodetection performance.

Figure 4f displays a comparison of the photocurrent properties with dielectric layer materials PTFE, polydimethylsiloxane (PDMS), and silicone rubber. Under similar UV illumination, the photocurrents of the IEFB-SP devices with PDMS, PTFE, and silicone rubber were 118.30 nA, 1.97 μA, and 0.04 nA, respectively. The response speeds of the devices with PDMS or with silicone rubber were slower than the self-powered photodetector with PTFE. Since the built-in electrostatic field is derived from the charge transfer between TiO_2_ and the dielectric materials, the ability to gain electrons from TiO_2_ plays a key role in the photodetection performance of the IEFB-SP photodetector. It is closely dependent on the electronegativity of materials. The better the electronegativity of a material is, the higher its ability to gain electrons from other material is, and the stronger the electrostatic field it yields. A stronger electrostatic field can be generated in the interface when a material of high electronegativity is in contact with a material of high electropositivity. In the triboelectric series [43,44], among these three materials, PTFE exhibited the strongest capability for gaining electrons and the highest electronegativity [45]. Therefore, the device with PTFE has the best photodetection performance.

### 3.5. Performance Comparison

Finally, Table 1 lists a comparison of state-of-the-art photodetectors and our proposed device. Our IEFB-SP photodetector exhibited excellent comprehensive detection performance, including high optical responsivity and sensitivity, especially with large photocurrent and ultra-low dark current. It is worth noting that we do not need a high-quality semiconductor junction for to achieve high-performance photodetection. Moreover, universal devices can be devised for other types of junctions to improve function. We provide a novel strategy for the development of high-performance, easy-to-use, and low-cost photodetectors.

## 4. Conclusions

In summary, the charge transfer characteristics between the TiO_2_ and the dielectric interfaces are quantitatively analyzed both from theoretical simulations and experiments. Furthermore, an electret of PTFE was inducted to remarkably boost the photovoltaic effect of a semiconductor heterojunction to develop self-powered UV photodectors. The proposed IEFB device achieved outstanding photodetection performances without a power source. Under the illumination of a xenon lamp, the optical responsivity of the IEFB device with PTFE was 76.87 mA/W, the specific detection rate was 4.79 × 10^12^ jones, and the EQE reached 26.48% at a wavelength of 360 nm. It also exhibited a rapid rise and decay time of 7.44 and 3.75 ms. Compared with a conventional heterojunction device under the same experimental conditions, the IEFB-SP photodetector demonstrated its photoresponse, light–dark ratio, rise speed, and decayed time improved by 442, 8.40 × 10^5^, 111, 267 times those of traditional device. Furthermore, the IEFB-SP device exhibited excellent temperature and circumstance stability. The photodetection performance remained unchanged from room temperature to 70 °C and did not degrade after 60 days of exposure under air surroundings without packaging. Overall, the IEFB-SP device achieves excellent photodetection performance without the requirement of high-quality junctions. It has the advantages of stable high-performance, simplicity, ease-of-fabrication, and low-cost. The strategy of using a built-in electrostatic field on the semiconductor–electret interface to modulate the photovoltaic effect is universal for various types of semiconductor junction devices. It is of significant guidance to improve the performance of self-powered photodetectors. The results in this work deepened the understanding of the electrostatic effect of the dielectric interface in the micro-region and are expected to inspire more applications in the future.

## Figures and Tables

**Figure 1 nanomaterials-12-03200-f001:**
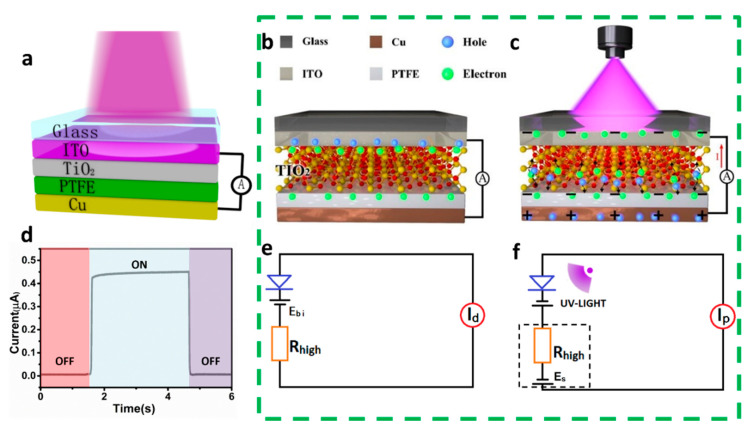
**The device structure and working mechanism.** (**a**) Design of the IEFB-SP ultraviolet photodetector. Working mechanism of the IEFB-SP device in light-off (**b**) and light-on (**c**) states. (**d**) Photocurrent of one switching cycle with 365-nm wavelength and illumination of 0.64-mW/cm2 at a bias of 0 V. Equivalent circuit diagram of the device in light-off (**e**) and light-on (**f**) states.

**Figure 2 nanomaterials-12-03200-f002:**
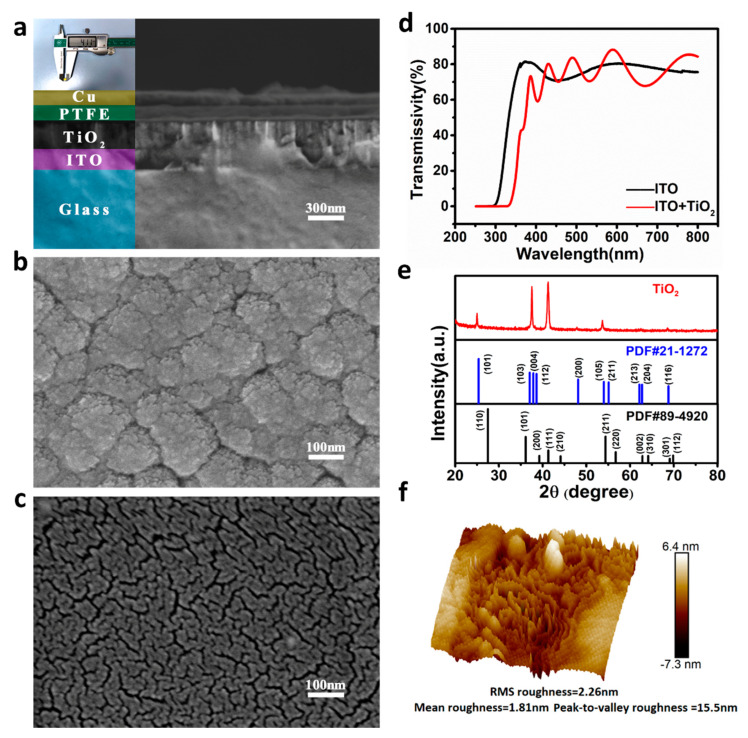
**Material characterization.** (**a**) Cross-sectional scanning electron microscopy (SEM) images of ITO/TiO_2_/PTFE/Cu film and photograph of the IEFB-SP photodetector. SEM images of TiO_2_ (**b**) and PTFE (**c**) surfaces. (**d**) Optical transmission spectra of the ITO with glass substrate and the sample after the growth of TiO_2_ film. (**e**) X-ray diffraction (XRD) pattern of the TiO_2_ film. (**f**) AFM image of the TiO_2_ surface.

**Figure 3 nanomaterials-12-03200-f003:**
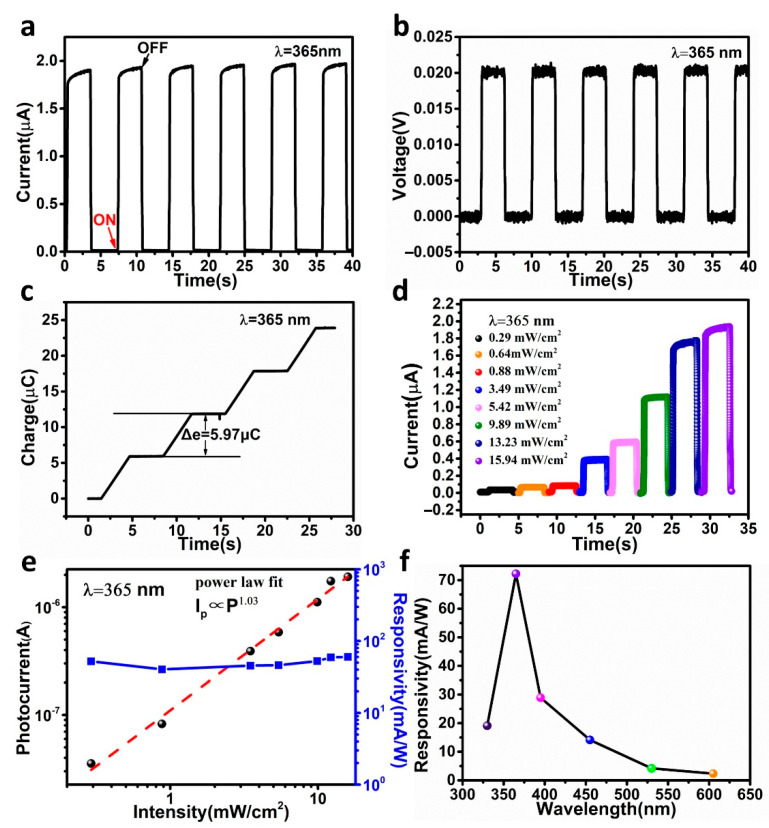
**Photodetection performance of IEFB-SP device.** Photocurrent (**a**) and photovoltage (**b**) under 365-nm UV light illumination (15.94 mW/cm^2^) at 0 V. (**c**) The number of transferred charges in each illumination. (**d**) Intensity-dependent photocurrent of the IEFB-SP device. (**e**) Photocurrent and responsivity as a function of illumination intensity. (**f**) Photoresponsivity at various wavelengths (325, 365, 396, 457, 532, and 607 nm).

**Figure 4 nanomaterials-12-03200-f004:**
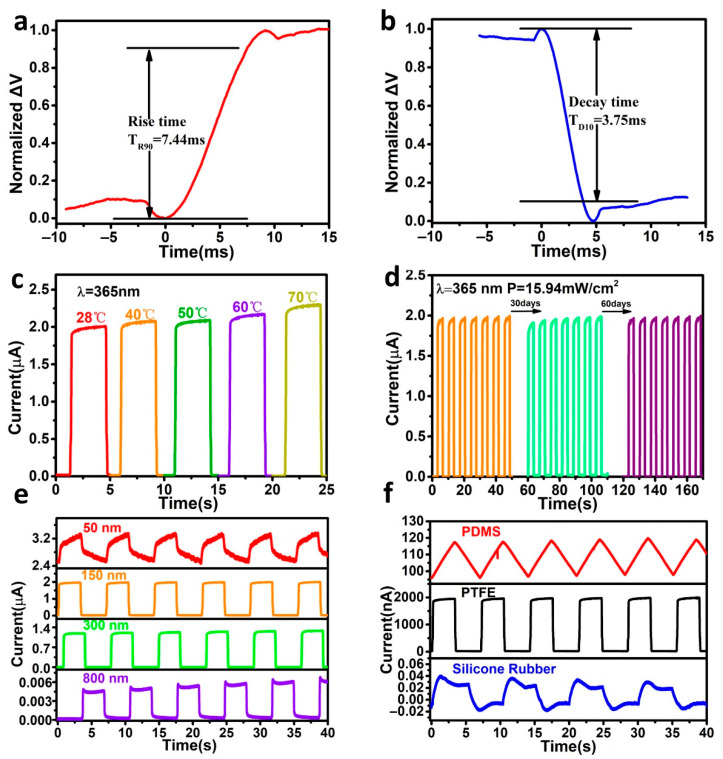
**Response speed and stability of the IEFB-SP photodetector with PTFE and dielectric material influence.** The rise time (**a**) and the decay time (**b**) of the IEFB-SP photodetector with PTFE; (**c**) Photocurrents of the IEFB-SP photodetector with PTFE at different temperatures; (**d**) Photocurrents of IEFB-SP photodetector with PTFE in air environment for 30 and 60 days; (**e**) Photocurrents of the device with different PTFE thicknesses (50, 150, 300, and 800 nm); (**f**) Photocurrents of IEFB-SP photodetector with different dielectric materials.

**Table 1 nanomaterials-12-03200-t001:** Comparation of self-powered UV photodetectors.

Materials	Type	Wavelength (nm)	Responsivity (mA/W)	I_p_ (μA)	I_d_ (μA)	Sensitivity	Rise Time (ms)	Decay Time (ms)	Ref.
**ZnO MSM**	**Schottky junction**	**365**	**20**	**–**	**–**	**–**	**7.1 × 10^−4^**	**4 × 10^−3^**	**[46]**
**FTO, TiO_2_, Ag NWs**		**200–400**	**32.5**	**1.72**	**1.01 × 10^−3^**	**1.7 × 10^3^**	**44 × 10^−6^**	**1.85 × 10^−3^**	**[47]**
**Au/β-Ga_2_O_3_**		**254**	**0.01**	**1.5 × 10^−4^**	**1 × 10^−5^**	**≈15**	**0.001**	**0.06**	**[48]**
**ZnO/graphene**		**380**	**39**	**–**	**–**	**–**	**0.037**	**0.33**	**[11]**
**TiO_2_/Ag NWs**		**350**	**1.1**	**4.2 × 10^−3^**	**2.7 × 10^−8^**	**1.54 × 10^5^**	**2**	**47**	**[21]**
**n-Ga_2_O_3_/n-ZnO**	**Heterojunction**	**251**	**9.7**	**6 × 10^−3^**	**1 × 10^−5^**	**>10^2^**	**0.1**	**0.9**	**[49]**
**TiO_2_/NiO**		**350**	**0.065**	**3.8 × 10^−4^**	**2.5 × 10^−5^**	**14**	**1200**	**7100**	**[50]**
**Cs_3_Cu_2_I_5_/β-Ga_2_O_3_**		**265**	**2.3**	**0.51**	**1 × 10^−5^**	**5.1 × 10^4^**	**37**	**45**	**[51]**
**Au/Spiro-** **OMeTAD/FeBHT/SnO_2_/ITO** **ITO/SnO_x_/** **MA_3_Bi_2_I_9_/Spiro-** **OMeTAD/Au**	**Perovskite**	**365** **382**	**6.57** **280**	**2.25** **10**	**–**	**–** **–**	**<40** **380**	**<40** **450**	**[52]** **[53]**
**Pt NP@TiO_2_/** **GaN NRs** **Ppy/GaN NRs** **Ag NWs @ Ppy-PEDOT:PSS/GaN NRs**	**Hybrid heterostructure**	**382** **382** **382**	**4.46 × 104** **1.02 × 10^5^** **3.1 × 10^6^**	**153** **1.02** **20**	**0.2** **0.10** **9.52 × 10^−3^**	**765** **9.80** **2.1 × 10^3^**	**180** **350** **200**	**200** **410** **210**	**[54]** **[55]** **[56]**
**ITO/TiO_2_/PTFE/Cu**	**IEFB Heterojunction**	**360**	**76.87**	**2.00**	**8.03 × 10^−6^**	**2.49 × 10^5^**	**7.44**	**3.75**	**our work**

## Data Availability

Not applicable.

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
