# Peer review of "An Internal-Electrostatic-Field-Boosted Self-Powered Ultraviolet Photodetector"

_nanomaterials, 2022, doi:10.3390/nano12183200_

Round 1

Author Response

Response to Reviewer 1 Comments

Thank you for your letter and the reviewer’s precious comments and suggestions concerning our manuscript. These comments and suggestions are not only very helpful for revising and improving our manuscript, but also of great guiding significance to our research. We have studied the comments and suggestions carefully and have made corrections which we hope meet with approval. The revised portions are marked in red in the manuscript. The main corrections in the manuscript and the responses to the reviewer’s comments and suggestions are as follows:

Point 1: In section 3.1 and Fig. 1, the Authors explained mechanisms of how to work the selfpowered UV photodetector. However, only from the sentences and equivalent circuits, readers (at least I) do not understand the mechanism well, especially interfacial electron/hole accumulation or separation. To grasp the working mechanism, I strongly suggest to drawing energy diagram of the TiO2/ITO and TiO2/PTFE interfaces for explaining the reason why current flows as shown in Fig. 1(c).

Response 1: Thank you for your valuable questions and suggestions. In our ESPB-SP device, PTFE, as a polar polymer material, plays two roles. First, it is used as a dielectric material, and after contact with TiO2, charge transfer occurs between the interface of TiO2 and PTFE, while the charge on the surface of PTFE is bound and cannot move freely, thus creating an internal electrostatic field between the interfaces. In the light-on state, TiO2 absorbs UV photons and generates electron-hole pairs. While on the contact surface of ITO and TiO2, a built-in electric field is generated in the heterojunction region due to the different work functions and carrier concentrations between the two semiconductor materials. Under the action of the built-in electric field, electron-hole pairs separate and diffuse and are collected by the electrode, forming a circuit outside. The built-in electrostatic field boosts the separation of electron-hole pairs and pushes the electrons to ITO while the holes to PTFE, as shown in Figure R1. With the coupling effect of the built-in electric field and the electrostatic field, a greatly enhanced photovoltaic effect is achieved. Second, it acts as an insulator to provide extremely high resistance to the device, which effectively reduces the dark current of the device. Thus, the enhanced effect of the electrostatic field maintains a high photocurrent and greatly reduces the dark current of the device, and in this dual effect, high-performance self-powered detection of the device is achieved. Given that PTFE is an insulator with a theoretically infinitely wide band gap, it is almost impossible for electrons to jump from the valence band to the conduction band, so we cannot map the energy diagram between the device interfaces.

Figure R1. Schematic diagram of photo-generated carriers transport under the coupling effect of built-in electric field (Ebi) and electrostatic field (Es).

Point 2: In Fig. 2(e), the Authors indicated rutile TiO2 from the XRD pattern. However, the peak positions are greatly different between the experiments and JCPDF card. I think that the TiO2 contained both rutile and anatase phases. The XRD peak at 2theta =25º  would be anatase (101). Please check a JCPDF card of anatase TiO2. In fact, the growth and annealing temperatures were not high. Thus, mixture of anatase and rutile phases is reasonable.

Response 2: Thank you for your valuable comment. As shown in Figure R2, we added anatase PDF#21-1272 card to the Fig. 2(e) in the manuscript, where peak (101) and peak (103) show the anatase phase of TiO2 prepared in this paper and peak (111) and peak (211) show the rutile phase of TiO2 prepared in this paper, and as stated by the reviewer, the TiO2 prepared in the paper is a mixture of anatase and rutile phases.

R2. X-ray diffraction (XRD) pattern of the TiO2 film PDF card of anatase and TiO2.

Point 3: In Fig. 3(c), the data of the transferred charges are shown. However, I do not know how to measure them. Are they really measured or simply converted from (photo) current? Please explain in more detail.

Response 3: Thank you for your good comment. The amount of transferred charge between two electrodes of ESPB-SP PD was measured by an electrometer (KEITHLEY 6514). The ESPB-SP PD with PTFE can be regarded as a power source with high internal resistance. The electrometer was directly connected with the positive and negative electrodes of PD and the amount of transferred charge in the circuit can be obtained by the charge measurement function of the electrometer. The charge transfer measured by the electrostatic meter in 3 seconds was 5.97 µC, while the photocurrent calculated by the equation I = Q/t was 1.99 × 10−6 A, which also corresponds to the current measurement.

Point 4: For Table 1, please revise it as horizontal format if nanomaterials journal accepts to use such table. Some words and numerical values are written across two lines in Table 1(Wavelength, Responsivity, 200–400, ...), which is hard to see.

Response 4: Thank you for your precious suggestion. We have revised Table 1 to make it clearer.

Point :The followings are minor comments.

  1. A) In abstract, please revise 24900000 % to 2.49 x 10^7 %.
  2. B) On Line 134, please revise first calculation principle to first principles calculation.
  3. C) Max roughness shown in Fig. 2 is usually represented as peak-to-valley (PV) roughness.
  4. D) On Line 202, 8.40 x 105 times →40 x 10^5 times.
  5. E) In Figs. 4(a) and 4(b), please revise the horizontal axis to ms or 10^-3 s.

Response: Thank you for your good suggestions. We have revised and marked in red in the manuscript.

Reviewer 2 Report

Comments to Author:

The manuscript can be accepted only after major revision is been ascertained, authors need to address these comments.

1.      Title: Revise the title of manuscript.

2.      Abstract: Recheck and correct the on-off ratio value should be present as in Line No 188.

3.      Rewrite the introduction part with more citations and explain more about your work structure, and its importance. (Ex. 10.1088/1361-6641/abda62

10.1016/j.snb.2020.129175, 10.1016/j.chroma.2022.462997)

4.      Insert the absorption results and band gap values from Tauc’s plots of studied samples.

5.      It should be good for readers if you include the XPS results of ITO/TiO2/Cu with and without the PTFE layer in this manuscript.

6.      According to author’s statement, why the photocurrent values are only high for PTFE thickness of 150 nm and explain?

7.      Authors provided the photocurrent results with PTFE in air environment and should be present photocurrent results under humidity condition also.

8.      Check Table 1 and the correct photocurrent and dark current units should be present in a similar range (i.e. nA or µA)

9.      In Table 1, check the rising time and decay time units (ms), it should be as “sec”.

Cite recent articles in introduction part and compare in Table.

10.1016/j.apsusc.2022.153474, 10.1002/pssr.202000518, 10.1002/adom.202100192, 10.1021/acsami.0c16795

10.  Replot the Fig. 4 (a) and 4 (b) and fit carefully.  

11.  Authors should explain the detailed mechanism using plausible band alignment in addition to the Figure S3. Since the charge transfer between TiO2/PTFE layer has been highlighted.

12.  Recheck the Fig. caption and description in entire manuscript. (ex. S2 was not described in manuscript).

13.  Overall the manuscript results are interesting. Should answer for all questions and do the required modifications/corrections for accepting the manuscript.

Author Response

Response to Reviewer 2 Comments

Thank you for your letter and the reviewer’s precious comments and suggestions concerning our manuscript. These comments and suggestions are not only very helpful for revising and improving our manuscript, but also of great guiding significance to our research. We have studied the comments and suggestions carefully and have made corrections which we hope meet with approval. The revised portions are marked in red in the manuscript. The main corrections in the manuscript and the responses to the reviewer’s comments and suggestions are as follows:

Point 1: Title: Revise the title of manuscript.

Response 1: Thank you for your precious suggestions. Our device is an innovative self-powered operating mode by generating an internal electrostatic field between the interface through contact between PTFE and TiO2 to modulate the photovoltaic effect of the ITO/TiO2 heterojunction, so we have revised the title of the manuscript to An Internal-Electrostatic-Field-Boosted Self-Powered Ultraviolet Photodetector.

Point 2: Abstract: Recheck and correct the on-off ratio value should be present as in Line No 188.

Response 2: Thank you for your good suggestion. We have revised “The self-powered UV photodetector with PTFE demonstrated an extremely high on-off ratio of 24900000%” in the abstract of the manuscript to “The self-powered UV photodetector with PTFE demonstrated an extremely high on-off ratio of 2.49×105”.

Point 3: Rewrite the introduction part with more citations and explain more about your work structure, and its importance. (Ex.10.1088/1361-6641/abda62 10.1016/j.snb.2020.129175, 10.1016/j.chroma.2022.462997)

Response 3: Thank you for giving good suggestions and providing references. We have added more explain of our detector work structure and cited several references you have provided.

Revision in the manuscript:

In this work, an electret was creatively used to enhance the photovoltaic effect and a self-powered UV photodetector with remarkable performances was fabricated. The internal-electrostatic-field-boosted self-powered (IEFB-SP) photodetector consisted of a semiconductor heterojunction and an electret layer. In our IEFB-SP PD, there exist two built-in electric fields. One is on the ITO/TiO2 interface and the other is on the TiO2/PTFE interface. ITO/TiO2 is known as a semiconductor heterojunction that produces a built-in electric field at the junction region due to the difference in the work function of the two materials. On the TiO2/PTFE interface, there exists a built-in electrostatic field because PTFE molecules with strong electronegativity gain electrons from TiO2 molecules. The electrostatic field generated by the negatively charged PTFE layer is in the same direction as the built-in electric field on ITO/TiO2 interface and the two electric fields are in series. Under UV light irradiation, TiO2 absorbs UV photons and generates electron-hole pairs. The electrostatic field boosts the separation of electron-hole pairs and pushes the electrons to ITO while the holes to PTFE. With the coupling effect of the built-in electric field and the electrostatic field, an enhanced photovoltaic effect is achieved.

References added in the revised manuscript are as follows:

  1. Reddeppa, M.; Park, B.; Pasupuleti, K. S.; Nam, D.; Kim, S.; Oh, J.; Kim, M., "Current–voltage characteristics and deep-level study of GaN nanorod Schottky-diode-based photodetector." Semicond Sci Tech 2021, 36, 035010.
  2. Reddeppa, M.; KimPhung, N. T.; Murali, G.; Pasupuleti, K. S.; Park, B.; In, I.; Kim, M., "Interaction activated interfacial charge transfer in 2D g-C3N4/GaN nanorods heterostructure for self-powered UV photodetector and room temperature NO2 gas sensor at ppb level." Sens. Actuators B Chem 2021, 329, 129175.
  3. Kumar, B. P.; Rao, P. V.; Hamieh, T.; Kim, C. W., "Comparative study of nitrogen doped multi walled carbon nanotubes grafted with carboxy methyl cellulose hybrid composite by inverse gas chromatography and its UV photo detectors application." J. Chromatogr. A 2022, 1670, 462997.

Point 4: Insert the absorption results and band gap values from Tauc’s plots of studied samples.

Response 4: Thank you for your precious suggestion. The absorption results of ITO and TiO2 are shown in Figure S5 (a), while S5 (b) and S5 (c) are the Tauc's plots analysis images of ITO and TiO2 respectively. Following the analysis of Tauc's plots, we can obtain a forbidden band width of 3.75eV for ITO and 3.43eV for TiO2, which we have also included in the Tauc's plots analysis image Insert Supplementary Information.

Figure S5. (a) Absorption results of the ITO with glass substrate and the sample after the growth of TiO2 film. (b) Band gap values of ITO from Tauc’s plots. (c) Band gap values of TiO2 from Tauc’s plots.

Point 5: It should be good for readers if you include the XPS results of ITO/TiO2/Cu with and without the PTFE layer in this manuscript.

Response 5: Thank you for your precious comment. Our device is mainly based on the electret nature of PTFE, which can generate an internal electrostatic field at the PTFE/TiO2 interface to accelerate the separation and diffusion of photogenerated carriers and enhance the photovoltaic effect of the ITO/TiO2 heterojunction. XPS is primarily used to test the elements and elemental quality of the shallow surface of the material. The materials we use are common and are grown layer by layer to form a film, and the elemental composition of the surface of each layer does not have a great impact on the performance of our devices, so we do not test the XPS of each layer of the device during the characterization stage.

Point 6: According to author’s statement, why the photocurrent values are only high for PTFE thickness of 150 nm and explain?

Response 6: Thank you for your good question. In our devices, the PTFE layer plays a dual role for the device. First, it is used as a dielectric material, and after frictional contact with TiO2, charge transfer will occur between the interface of TiO2 and PTFE, charge transfer occurs between the interface of TiO2 and PTFE, while the charge on the surface of PTFE is bound and cannot move freely, thus creating an internal electrostatic field between the interfaces, accelerating the separation and diffusion of photogenerated carriers and enhancing the photovoltaic effect of ITO/TiO2 heterojunction. Second, it acts as an insulator providing great resistance for the device, which can effectively reduce the dark current of the device. As shown in Figure 4e, the thicker the PTFE thickness, the lower the photocurrent and dark current. This is because thicker PTFE films greatly increase the internal resistance of the device, which leads to lower photodark currents. As shown in Figure 4f, the voltage becomes progressively larger as the thickness increases, indicating that thicker dielectric layers can store more charge to enhance the built-in electrostatic field. There is a competitive relationship between the internal resistance of the device and the magnitude of the built-in electrostatic field, and there is an optimum value that allows the device to maintain both a high photocurrent and a low dark current. Of the four thicknesses of PTFE film, the detector with a thickness of 150 nm shows the best performance of the detector.

Point 7: Authors provided the photocurrent results with PTFE in air environment and should be present photocurrent results under humidity condition also.

Response 7: Thank you for your good suggestion. The humidity at the time of our tests was between 30RH and 40RH, as indicated in the manuscript and marked in red.

Point 8: Check Table 1 and the correct photocurrent and dark current units should be present in a similar range (i.e. nA or µA)

Response 8: Thank you for your good suggestion. We have standardized the units of photocurrent and dark current in Table 1 to μA.

Point 9: In Table 1, check the rising time and decay time units (ms), it should be as “sec”.

Cite recent articles in introduction part and compare in Table. 10.1016/j.apsusc.2022.153474, 10.1002/pssr.202000518, 10.1002/adom.202100192, 10.1021/acsami.0c16795

Response 9: Thank you for your good comment. Due to the response speed of the device being at the ms level, the rise and decay times are fast and therefore we have standardized on “ms”. We have also cited several references you have provided in the introduction section and compared them in the table 1.

References added in the revised manuscript are as follows:

  1. Vuong, V.; Pammi, S. V. N.; Pasupuleti, K. S.; Hu, W.; Tran, V. D.; Jung, J. S.; Kim, M.; Pecunia,V.; Yoon, S. G., "Engineering Chemical Vapor Deposition for Lead-Free Perovskite-Inspired MA3Bi2I9 Self-Powered Photodetectors with High Performance and Stability." Adv. Optical Mater. 2021, 9, 2100192.
  2. Pasupuleti, K. S.; Chougule, S. S.; Jung, N.; Yu, Y.; Oh, J.; Kim, M., "Plasmonic Pt nanoparticles triggered efficient charge separation in TiO2/GaN NRs hybrid heterojunction for the high performance self-powered UV photodetectors." Appl. Surf. Sci 2022, 594, 153474.
  3. Pasupuleti, K. S.; Reddeppa, M.; Park, B.; Oh, J.; Kim, S.; Kim, M., "Efficient Charge Separation in Polypyrrole/GaN-NanorodBased Hybrid Heterojunctions for High-Performance Self-Powered UV Photodetection." Phys. Status Solidi RRL 2021, 15, 2000518.
  4. Pasupuleti, K. S.; Reddeppa, M.; Park, B.; Peta, K. R.; Oh, J.; Kim, S.; Kim, M., "Ag Nanowire-Plasmonic-Assisted Charge Separation in Hybrid Heterojunctions of Ppy-PEDOT:PSS/GaN Nanorods for Enhanced UV Photodetection." ACS Appl. Mater. Interfaces 2020, 17.

Point 10: Replot the Fig. 4 (a) and 4 (b) and fit carefully.

Response 10: Thank you for your good suggestion. We have replotted and carefully fitted Figs. 4 (a) and 4 (b), which have been revised and marked in red in the manuscript.

Point 11: Authors should explain the detailed mechanism using plausible band alignment in addition to the Figure S3. Since the charge transfer between TiO2/PTFE layer has been highlighted.

Response 11: Thank you for your precious suggestion. In our IEFB-SP device, PTFE, as a polar polymer material, plays two roles. First, it is used as a dielectric material, and after contact with TiO2, charge transfer occurs between the interface of TiO2 and PTFE, while the charge on the surface of PTFE is bound and cannot move freely, thus creating an internal electrostatic field between the interfaces. In the light-on state, TiO2 absorbs UV photons and generates electron-hole pairs. While on the contact surface of ITO and TiO2, a built-in electric field is generated in the heterojunction region due to the different work functions and carrier concentrations between the two semiconductor materials. Under the action of the built-in electric field, electron-hole pairs separate and diffuse and are collected by the electrode, forming a circuit outside. The built-in electrostatic field boosts the separation of electron-hole pairs and pushes the electrons to ITO while the holes to PTFE. With the coupling effect of the built-in electric field and the electrostatic field, a greatly enhanced photovoltaic effect is achieved. Second, it acts as an insulator to provides extremely high resistance to the device, which effectively reduces the dark current of the device. Thus, the enhanced effect of the electrostatic field maintains a high photocurrent and greatly reduces the dark current of the device, and in this dual effect, high-performance self-powered detection of the device is achieved. Given that PTFE is an insulator with a theoretically infinitely wide band gap, it is almost impossible for electrons to jump from the valence band to the conduction band, so we cannot map the plausible band alignment between the device interfaces.

Point 12: Recheck the Fig. caption and description in entire manuscript. (ex. S2 was not described in manuscript).

Response 12: Thank you for your good comment. We have rechecked the manuscript and added a description of S2, S3 and S4.
